# Neuropathic Pain Diagnosis Simulator for Causal Discovery Algorithm Evaluation

**Ruibo Tu**
KTH Royal Institute of Technology
ruibo@kth.se

**Kun Zhang**
Carnegie Mellon University
kunz1@cmu.edu

**Bo Christer Bertilson**
Karolinska Institute
bo.bertilson@ki.se

**Hedvig Kjellström**
KTH Royal Institute of Technology
hedvig@kth.se

**Cheng Zhang**
Microsoft Research, Cambridge
Cheng.Zhang@microsoft.com

## Abstract

Discovery of causal relations from observational data is essential for many disciplines of science and real-world applications. However, unlike other machine learning algorithms, whose development has been greatly fostered by a large amount of available benchmark datasets, causal discovery algorithms are notoriously difficult to be systematically evaluated because few datasets with known ground-truth causal relations are available. In this work, we handle the problem of evaluating causal discovery algorithms by building a flexible simulator in the medical setting. We develop a neuropathic pain diagnosis simulator, inspired by the fact that the biological processes of neuropathic pathophysiology are well studied with well-understood causal influences. Our simulator exploits the causal graph of the neuropathic pain pathology and its parameters in the generator are estimated from real-life patient cases. We show that the data generated from our simulator have similar statistics as real-world data. As a clear advantage, the simulator can produce infinite samples without jeopardizing the privacy of real-world patients. Our simulator provides a natural tool for evaluating various types of causal discovery algorithms, including those to deal with practical issues in causal discovery, such as unknown confounders, selection bias, and missing data. Using our simulator, we have evaluated extensively causal discovery algorithms under various settings.

## 1 Introduction

Many real-life decision-making processes require an understanding of underlying causal relations. For example, understanding the cause of symptoms is essential for physicians to make correct treatment decisions; understanding the cause of observed environmental changes is critical to take action against global warming. However, it is generally infeasible or even impossible to do interventions or randomized experiments to verify these causal relations. Therefore, causal discovery from observational data has attracted much attention [29, 31, 40, 49].

However, the evaluation of causal discovery algorithms has been a challenge [3]. The great application demand also indicates that ground-truth causal relations in a complex scenario are often unknown to humans. The lack of systematic evaluations of causal discovery algorithms has hindered both the development of the field and the impact of these algorithms on solving real-life problems. Research-wise, it is hard to identify the advantages and disadvantages of causal discovery algorithms performing in real-world scenarios. A systematic way to evaluate causal discovery algorithms is pressing.

Other machine learning disciplines such as supervised learning and reinforcement learning have made great success in real-world applications such as image classification [34, 45] and speech recognition [2]. An important driving factor for their fast development and great success is the existence of a large amount of benchmark datasets for systematic evaluation. The benchmark datasets can be in the form of large-scale labeled and publicly available datasets such as [13, 22], which are commonly used for supervised and unsupervised learning. They can also be in the form of synthetic data that are generated from simulators, e.g. an autonomous driving simulator [4], an agent motion [5], and a gaming environment [19]. Such simulators accelerate the development of reinforcement learning algorithms and promote usage in real-life applications.

Establishing benchmark datasets for the evaluation of causal discovery algorithms will naturally accelerate the development of this research discipline and increase its real-world impact. However, it is difficult to collect such datasets with known ground-truth because underlying real-world causal relations are usually highly complex. Fortunately, domain experts in disciplines such as biology and physics can provide information about well-understood causal influences in some specific scenarios. This gives us opportunities to utilize domain knowledge to reveal ground-truth causal relations and build realistic simulators. In this way, we can generate data from simulators and use such benchmark datasets for the evaluation of causal discovery algorithms.

In this work, we present a neuropathic pain diagnosis simulator for evaluating causal discovery algorithms. As one of the most important healthcare issues, neuropathic pain is well-studied in bio-medicine and has well-understood causal influences. By definition, neuropathic pain is caused by disease or injury of the nervous system. It includes various chronic conditions that, together, affect up to $8\%$ of the population. The prevalence of neuropathic pain increased to $60\%$ in those with severe clinical neuropathy [9]. We build a simulator based on the causal relations in neuropathic pain diagnoses. Given the causal relations, we estimate the parameters of the corresponding causal graph using a small cohort of anonymous real-world clinical records to generate simulated data. Our simulator not only provides the simulated data and the ground-truth causal relations for evaluating causal discovery algorithms but also builds up a bridge between machine learning and neuropathic pain diagnoses. In summary, our contribution is a neuropathic pain diagnosis simulator. Especially:

- It represents a complex real-world scenario with more than 200 variables and around 800 well-defined causal relations. It can also generate any amount of data without jeopardizing security or privacy of patients' data (Section 2).

- Our simulator can produce data indistinguishable from real-world data. We have verified the simulation quality using both medical expertise and statistical evaluation (Section 3).

- Our simulator is flexible and can be used to generate data with different practical issues, such as confounding, selection bias, and missing data (Section 2.3 and Section 4).

- We have evaluated major causal discovery algorithms, including PC [40], Fast Causal Inference (FCI) [40], and Greedy Equivalence Search (GES) [6] with simulated data under different settings (Section 4).

## 2 Neuropathic Pain Simulator

In this section, we introduce our neuropathic pain diagnosis simulator.[1] We first show essential causal relations in the neuropathic pain diagnosis, and then present details of the simulator design. Finally, we discuss some open problems in causal discovery and how to use our simulator to simulate instances of such problems.

### 2.1 Causal Relations for Neuropathic Pain Diagnosis

Neuropathic pain diagnoses mainly contain symptom diagnosis, pattern diagnosis, and pathophysiological diagnosis. For example, Table 1a shows typical neuropathic pain diagnostic records. *Symptom diagnosis* describes the discomfort of patients. *Pattern diagnosis* identifies symptom patterns. In neuropathic pain diagnosis, it identifies which set of nerves do not work properly. Such conditional is commonly called Radiculopathy. The main tool of pattern diagnosis is the dermatome map as shown in Figure 1. *Pathophysiological diagnosis* refers to the original cause of symptoms where

Table 1: Diagnostic records and dataset.

(a) A typical neuropathic pain diagnostic record. "L" and "R" stand for "left" and "right".

| |
|---|
| **Symptom diagnosis:** R back thigh discomfort, R knee discomfort, L knee thigh discomfort, Patellofemoral pain syndrome |
| **Pattern diagnosis:** L L5 Radiculopathy, R L5 Radiculopathy |
| **Pathophysiological diagnosis:** Discoligment injury L4-5 |

(b) Given many patient records, a diagnostic record dataset takes the following form. "ID" represents different patients. "DLI" and "Radi" stand for discoligamentous injury and radiculopathy. Each row is a patient's diagnostic record in which "1" represents that the patient has the symptom and "0" represents that the patient has no such symptom.

| ID | DLI C1-C2 | DLI C2-C3 | ... | L C5 Radi | ... | R knee | L neck | ... |
|----|-----------|-----------|-----|-----------|-----|--------|--------|-----|
| 1  | 0         | 0         | ... | 1         | ... | 1      | 0      | ... |
| 2  | 1         | 0         | ... | 0         | ... | 0      | 1      | ... |
| ...| ...       | ...       | ... | ...       | ... | ...    | ...    | ... |
| n  | 0         | 1         | ... | 0         | ... | 0      | 0      | ... |

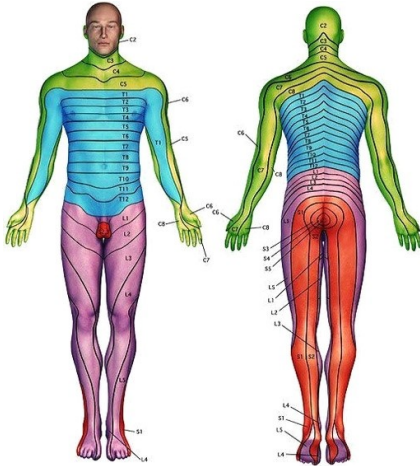

Figure 1: Dermatome map (image source [1]) shows surface regions of different nerves.

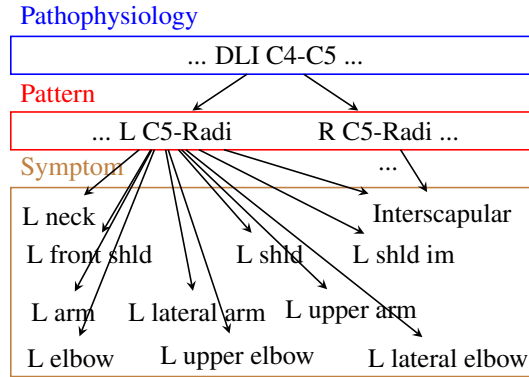

Figure 2: Typical structure of the ground-truth causal graph. "DLI" and "Radi" represent discoligamentous injury and radiculopathy. "shldr" and "im" stand for shoulder and impingement. "L" and "R" stand for left and right. We show the left side symptoms, and the corresponding connections are the same on the right side.

discoligamentous injury is the most common factor in the neuropathic pathophysiological diagnosis. Given a set of patient data, we can present the data as in Table 1b, where 1 indicates that the diagnostic label exists in a patient record and 0 otherwise.

In neuropathic pain diagnoses causal relations are well studied in biomedical research [27, 43]. In general, neuropathic pain symptoms in symptom diagnosis are mainly caused by radiculopathies (Radi) in the pattern diagnosis, and the radiculopathy is mostly caused by discoligamentous injuries (DLI) in the pathophysiological diagnosis. For example, some of the causal relations are shown in Figure 2. DLI C4-C5 causes left side C5 radiculopathy and right side C5 radiculopathy. Left side C5 radiculopathy further causes symptoms at the left front shoulder, the left lateral arm, etc. We see that these locations are consistent with the dermatome map in Figure 1. Despite that there are other causes of neuropathic pain symptoms and radiculopathies such as tumors and diabetes, they rarely appear in primary care. Therefore, we focus on the causal relations among the discoligamentous injuries, radiculopathies, and neuropathic pain symptoms in this work.

The complete causal relations are summarized in Appendix A, and we further provide interactive causal graph visualization at: https://cutt.ly/BekNFSy. The causal graph is similar to Figure 2

and consists of three layers: Symptom diagnosis, pattern diagnosis, and pathophysiological diagnosis. Nodes in each layer have no connection with each other. Arrows either point from nodes in the pathophysiological diagnosis layer to nodes in the pattern diagnosis layer or from nodes in the pattern diagnosis layer to nodes in the symptom diagnosis layer. The causal graph also contains different d-separations such as the folk structure, denoted by $\wedge$ structure (e.g., Left C5 Radiculopathy $\leftarrow$ Discoligamentous injury C4-C5 $\rightarrow$ Right C5 Radiculopathy), the collider structure, denoted by $\vee$ structure (e.g., Left C5 Radiculopathy $\rightarrow$ Left neck pain $\leftarrow$ Left C4 Radiculopathy), and the chain structure (e.g., Discoligamentous injury C4-C5 $\rightarrow$ Left C5 Radiculopathy $\rightarrow$ Left Neck pain).

## 2.2 Neuropathic Pain Diagnosis Simulator

With the domain knowledge mentioned in Section 2.1, we create our simulator to generate patient diagnostic records.

**Real-world diagnostic records.** To make our generated records close to the real-world scenario, we learn parameters from a dataset including 141 patient diagnostic records [46]. [2] These patients' diagnostic records are represented as a table of binary variables as shown in Table 1b. The variables in the pathophysiological diagnosis consist of the craniocervical junction injury and 26 discoligamentous injuries; the variables in the pattern diagnosis include 52 radiculopathies; the variables in the symptom diagnosis contain 143 symptoms. Similar to the real-world diagnostic records, the columns of generated records are the mentioned variables and the rows represent the synthetic patients.

**Parameter estimation of the causal graph.** We estimate the Conditional Probability Distribution (CPD) of each variable given its parents in the causal graph with the real dataset. We compute the CPD of a variable $X$ by $P(X \mid Pa(X)) = \frac{P(X, Pa(X))}{P(Pa(X))}$, where $Pa(X)$ represents the parents of $X$ in the causal graph. Since variables are binary, the joint distributions can be computed using the number of variable values in the dataset. However, we cannot estimate the CPDs accurately for the variables with many parents because of the curse of dimensionality and the limited number of the real data. Therefore, instead of computing the CPD of $X$ given all its parents, we introduce the heuristic

$$P(X = 1 \mid Pa(X) = \mathbf{c}) \leftarrow \max_{i \in I_1} P(X = 1 \mid Pa_i(X) = c_i), \qquad (1)$$

where $\mathbf{c}$ is a given vector of parent values (which can contain either value zero or one), and $I_1$ is a subset of the index of all variables in $Pa(X)$ such that for $\forall i \in I_1$, $Pa_i(X) \in Pa(X)$ and $c_i = 1$. The condition of Equation 1 is that there exists $c_i \in \mathbf{c}$ such that $c_i = 1$. This condition is satisfied in the real data. Given the parent values $\mathbf{c}$, we only consider the parents taking the value one, and get the maximum conditional probability of $X = 1$ given a parent taking the value one in $\mathbf{c}$ to estimate the CPD of $P(X = 1 \mid Pa(X) = \mathbf{c})$.

This approximation is supported by the medical insights. Intuitively, if a symptom is caused by multiple nerves, the chance for the symptom to exist in general is higher when these causes occur at the same time comparing to only one of the causes occurs. For example, both $L4$ and $L5$ radiculopathies can cause knee pain. The chance that a person with both $L4$ and $L5$ radiculopathies feels knee pain is higher or equal to the chance that a person with either one of the radiculopathies feels knee pain. In other words, $P(X = 1 \mid Pa_1(X) = 1, Pa_2(X) = 1) \geq P(X = 1 \mid Pa_1(X) = 1)$ and $P(X = 1 \mid Pa_1(X) = 1, Pa_2(X) = 1) \geq P(X = 1 \mid Pa_2(X) = 1)$, where $Pa_1(X)$ and $Pa_2(X)$ are $L4$ and $L5$ radiculopathies and $X$ is knee pain.

Given all the conditional probability and marginal probability distributions, we use ancestral sampling to sample neuropathic pain diagnosis data of synthetic patients.

## 2.3 Simulating Data with Practical Issues of Causal Discovery

Causal discovery is facing many practical issues when applied in real-world applications. Our simulator has many advantages over real datasets in evaluating causal discovery algorithms in the presence of these challenges. In this section, we introduce how to use our simulator to generate

datasets exhibiting different open problems. In Section 4 we show experimental results of applying causal discovery algorithms to these simulated data reflecting different real-world problems.

**Unmeasured Confounding.** Most causal discovery algorithms assume that all variables of concerned are observed. However, in most real-life applications collected datasets may not cover all factors to discover causal relations of interest. If there is an unobserved common direct cause of two or more observed variables, this may produce wrong causal conclusions. This problem is known as unmeasured confounding, which is one of the common issues that one is faced with when applying causal discovery algorithms. Addressing unmeasured confounding is an active research direction [18, 20, 28, 40, 47].

There are many ways for our simulator to generate datasets of unmeasured confounding. We can delete the data of parent nodes in a ∧ structure. More specifically, deleting the simulated data of the pathophysiology diagnosis and the pattern diagnosis variables leads to confounding in the dataset because they have at least two direct effects. We can also introduce external variables as confounders in the data generation process. For example, we can add patients' occupation as a confounder which is not included in the given causal graph. The occupation affects daily activities and then increases the risk level of injuring different spine parts. With such datasets, we can evaluate how unmeasured confounding influences the results of causal discovery algorithms and hopefully develop new and better algorithms to address this issue.

**Selection bias.** Selection bias is an important issue in learning causal structures from real-world observational data. In practice, it is a common scenario where the data collection process is influenced by some attributes of variables. For example, samples in a dataset are not drawn randomly from the population, but from the people who have higher degrees than a bachelor's degree. Then, the selection variable is whether a person has a higher degree than a bachelor's degree. Such selection bias is non-trivial to be removed from the collected dataset and may introduce erroneous causal relations in the results of causal discovery algorithms. Few methods have been developed to address this issue [11, 12, 39, 47, 48]. We can also introduce selection bias to the simulated data. We first choose variables which the selection depends on, and then remove or maintain records based on the values of the chosen variables in the simulated dataset.

**Missing data.** Missing data is a ubiquitous issue, especially in healthcare. It is common to classify missingness mechanisms into Missing Completely At Random (MCAR), Missing At Random (MAR), and Missing Not At Random (MNAR) [32]. Among them, MAR and MNAR may introduce wrong causal conclusions if one simply deletes the data with missing entries, and applies causal discovery algorithms to the deleted complete dataset. Thus, methods that can handle different missingness mechanisms are much in demand for causal discovery [23, 24, 38, 42, 44].

Using our simulator, we can easily generate data with different missingness mechanisms. We can introduce missingness indicators to our causal graph. We then introduce causal relations between missingness indicators and substantive variables, depending on the missingness mechanism wanted. In the end, we sample the missingness indicators and mask out the data according to the values of missingness indicators.

## 3 Simulation Quality

We now evaluate whether generated data from our simulator have the similar property to the real-world data. We examine the quality of our simulated data by medical experts and statistical analysis.

### 3.1 Physician Evaluation

To examine the quality of our simulated data, we mix 50 simulated records with 50 records sampled from the real-world dataset. We then ask a physician specialized in neuropathic pain diagnoses to rate the 100 mixed records with the following score system:

- Score 1: This is not likely to be a real patient (possible but never see such patient before);
- Score 2: This is likely to be a real patient but is not very common (similar cases have happened before but rarely);

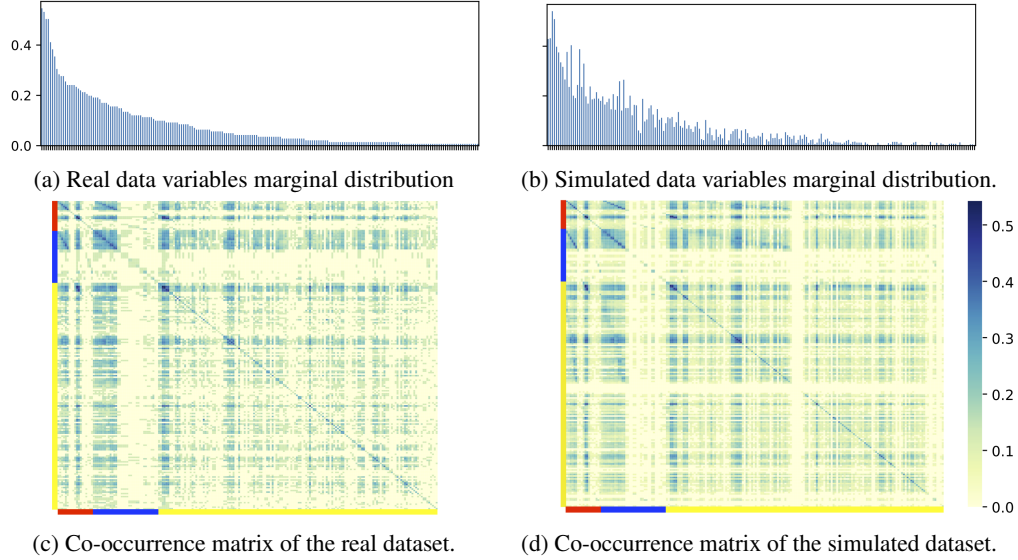

(a) Real data variables marginal distribution

(b) Simulated data variables marginal distribution.

(c) Co-occurrence matrix of the real dataset.

(d) Co-occurrence matrix of the simulated dataset.

Figure 4: Comparison of the marginal distributions and the co-occurrence matrices of the real and simulated datasets. The orders of variables are the same in Panel (a) and (b). In Panel (c) and (d), the red color represents pathophysiological diagnosis, the blue color represents pattern diagnosis, and the yellow color represents symptom diagnosis.

- Score 3: This is a common patient (similar cases show up time by time);
- Score 4: This is a typical patient (similar cases show up very often).

The physician evaluates the 100 records without knowing the source of the records (the simulator or the real dataset). Figure 3 shows the physician's evaluation results of the real and the synthetic data. The number of records with higher scores is increasing with the synthetic data which is expected due to our score design. The simulator generates less unlikely diagnostic records than those in the real datasets, which may be due to the missing and noisy labels in the real-world data. Also, when one or two unlikely diagnostic records are generated within many likely diagnostic labels in a record, the physician considers the case as "likely". This case happens more in the simulated data than the real-world data. In general, the result shows that the physician cannot differ the generated data from the real-world data. Also, the simulated data follow the desired frequency (increased numbers for higher scores) from the physician evaluation.

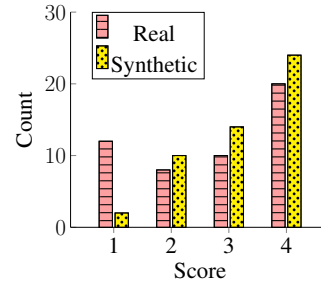

Figure 3: Physician's evaluation results of 50 real data and 50 simulated data.

## 3.2 Data Properties

We compare the marginal probability distributions of the same variables in the real dataset and the simulated dataset as shown in Figure 4a and Figure 4b. It shows that marginal probability distributions of variables in both datasets are similar.

We use the co-occurrence matrix normalized by the sample size to show the relation between each pair of variables in Figure 4c and Figure 4d. For example, the upper left corner of the co-occurrence matrices represents the relations between the variables in the pathophysiological diagnosis and the pattern diagnosis. We find that the pattern of the simulated data is similar to that of the real data. In our simulator, we give no constraints on the relations between both sides of variables, e.g. it is possible to have a connection between left $C5$ radiculopathy and right neck pain in the graph. We also compare the correlation matrices in Appendix B.

Table 2: Results of causal discovery algorithms using the real dataset and the simulated dataset with the same sample size. "CauAcc" and "Sim" represent "Causal Accuracy" and "Simulated".

| | CauAcc | | | | F1 | | Recall | | Precision | |
| | PC | GES | FCI | RFCI | PC | GES | PC | GES | PC | GES |
|---|---|---|---|---|---|---|---|---|---|---|
| Real | 0.041 | 0.038 | 0.024 | 0.021 | 0.044 | 0.037 | 0.025 | 0.022 | 0.187 | 0.199 |
| Sim | 0.038 | 0.063 | 0.023 | 0.016 | 0.047 | 0.076 | 0.025 | 0.043 | 0.425 | 0.377 |

Table 3: Results of different causal discovery algorithms with different sample sizes. The performance is better when causal accuracy and F1 score have larger values.

| Sample size | 128 | 256 | 512 | 1024 | 2048 | 4096 | 8192 | 16384 |
|---|---|---|---|---|---|---|---|---|
| $F1_{PC}$ | 0.019 | 0.028 | 0.016 | 0.040 | 0.066 | 0.100 | 0.142 | 0.188 |
| $F1_{GES}$ | 0.042 | 0.083 | 0.120 | 0.150 | 0.173 | 0.217 | 0.261 | 0.325 |
| $CauAcc_{PC}$ | 0.009 | 0.012 | 0.009 | 0.020 | 0.031 | 0.048 | 0.066 | 0.094 |
| $CauAcc_{GES}$ | 0.020 | 0.045 | 0.067 | 0.085 | 0.105 | 0.134 | 0.162 | 0.230 |
| $CauAcc_{RFCI}$ | 0.021 | 0.023 | 0.027 | 0.033 | 0.036 | 0.041 | 0.053 | 0.070 |
| $CauAcc_{FCI}$ | 0.026 | 0.029 | 0.034 | 0.039 | 0.045 | 0.051 | 0.062 | 0.082 |

## 4 Evaluating Causal Discovery Algorithms with Proposed Simulator

We evaluate major causal discovery algorithms with datasets generated from our simulator. We first further evaluate the simulation quality by comparing the causal discovery results of baseline methods between a real-world dataset and a simulated dataset. One advantage of the simulator is that we can generate any amount of data. Thus, we can evaluate causal discovery algorithms with different sample sizes to show the asymptotic property of causal discovery algorithms. Next, we apply causal discovery algorithms to the simulated datasets with different practical issues: Unmeasured confounding, selection bias, and missing data.

We use the causal discovery algorithms implemented by Tetrad [41]. In the experiments the causal discovery algorithms comprise: Constraint-based methods, PC [40], FCI [40], and RFCI [10]; score-based method, GES [6]. PC and GES output Complete Partially Directed Acyclic Graph (CPDAG), while FCI and RFCI output Partial Ancestral Graph (PAG). We use the F1 score and causal accuracy [7] as the evaluation metrics. Results of other metrics such as Structural Hamming Distance (SHD), precision, and recall are shown in Appendix .

**Comparison between simulated and real data.**    We sample 141 patient records from our simulator with the same sample size as the real-world dataset. We apply causal discovery algorithms to both datasets. The results are shown in Table 2. We find that the causal accuracies and F1 scores of both datasets are similar and the algorithms in the table cannot recover most edges of the ground-truth causal graph. The reason might be that the real dataset has a small sample size 141 compared with the number of nodes and edges in the causal graph. Moreover, Figure 4a shows that the appearance frequencies of diagnostic labels in the real dataset decay exponentially, which means that many diagnostic labels only appear in few patient diagnostic records. This is especially difficult for these methods because they are based on conditional independence tests that require sufficient samples. Furthermore, we find that the recall rates of PC on both datasets are similar and the precision rate of PC on the simulated dataset is higher than the precision rate on the real dataset. The reason might be that we generate values of a variable only based on the values of its parents. Consequently, our simulator can cancel out the influence of unknown confounders, such as the age and the occupation of a patient, and other practical issues in the real dataset. We also find that GES benefits relatively more than other methods from such property of the simulated dataset.

**Sample size.**    To show the influence of the sample size, we generate simulated datasets with sample size 128, 256, 512, 1024, 2048, 4096, 8192, and 16384. Under certain assumptions, these methods are asymptotically correct when infinite data are available. Table 3 shows that the performance of the algorithms is improved with increasing the sample size, when there is no selection bias, unknown confounders, or missing values. However, all these methods are not sample efficient as the F1 score and causal accuracy are still low and have not saturated even with 16834 data points. Thus, developing sample efficient causal discovery algorithms is needed, especially when real-life data are costly.

**Confounding.** We generate simulated data with external variables as confounders (see Appendix C for details). We compare the performance of FCI and RFCI on the dataset containing unknown confounders with that without confounders. The sample size of both datasets is 1024. The causal accuracy is 0.033 and 0.030 on the dataset with unknown confounders, and 0.039 and 0.033 on the dataset without unknown confounders. The results of the FCI algorithms on the dataset with unknown confounders are slightly worse than that without unknown confounders because the FCI algorithms consider the unknown confounders and output Partial Ancestral Graph (PAG) that provides the information about potential unknown confounders. However, it is far from ideal. We also generate confounding data by deleting all the data of some common parents in the causal graph. The results are shown in Appendix C.

**Selection bias.** We choose both sides of $C6$, $C7$, $L5$, and $S1$ radiculopathy as the causes of a selection variable. We then delete the simulated data regarding the values of the selection variable. We interpret this setting as a situation where the patients without those radiculopathies hardly ever go to the hospital; thus, the hospital hardly collects their data. Table 4 shows the results on the dataset with selection bias and the reference one without selection bias. RFCI is more robust to selection bias than FCI, even both should be able to handle it by design. For the algorithms without considering selection bias, the causal accuracy of GES outperforms PC.

Table 4: Results of different causal discovery methods in the presence of selection bias.

|  | FCI | RFCI | PC | GES |
|---|---|---|---|---|
| CauAcc | 0.039 | 0.039 | 0.031 | 0.109 |
| CauAcc$_\text{ref}$ | 0.046 | 0.037 | 0.033 | 0.114 |

**Missing data.** We evaluated the performance on all three missingness mechanisms: MCAR, MAR, and MNAR. We generate missing values in the dataset according to the definition in [23]. To generate the data that are MCAR, the probability distribution of missing values follows the Bernoulli distribution with the missingness probability 0.0007. To generate the data that are MAR, we choose variables in the pattern diagnosis as the causes of missingness indicators and variables in the pathophysiological diagnosis and the symptom diagnosis as the variables with missing values. Likewise, to generate the data that are MNAR, the variables with missing values are chosen in the range of all the variables in the causal graph. Since FCI, PC, and GES cannot deal with the dataset containing missing values, we delete the records containing any missing value and input the remaining complete dataset. The sample size of the remaining complete dataset is 7042. As a reference, we create a simulated dataset whose sample size is 7042 without missing values.

Table 5 shows that the results of MAR and MNAR experiments are worse than the results of MCAR experiments, which are close to the reference one without missing values. This is expected as [44] shows: When the data are MCAR, causal discovery results are asymptotically correct; when the data are MAR or MNAR, these algorithms may produce erroneous edges in the case where the missingness indicators are the common children or descendants of the common children of the concerned variables. We further check the number of missingness indicators satisfying this conclusion: 4 in MNAR and 7 in MAR out of total 52 missingness indicators.

Table 5: Results of applying causal discovery algorithms to the MCAR, MAR, and MNAR datasets.

|  | FCI | RFCI | PC | GES |
|---|---|---|---|---|
| CauAcc$_\text{MNAR}$ | 0.059 | 0.051 | 0.061 | 0.154 |
| CauAcc$_\text{MAR}$ | 0.063 | 0.049 | 0.050 | 0.135 |
| CauAcc$_\text{MCAR}$ | 0.066 | 0.055 | 0.067 | 0.161 |
| CauAcc$_\text{ref}$ | 0.062 | 0.050 | 0.059 | 0.145 |
| F1$_\text{MNAR}$ | X | X | 0.133 | 0.251 |
| F1$_\text{MAR}$ | X | X | 0.132 | 0.241 |
| F1$_\text{MCAR}$ | X | X | 0.141 | 0.256 |
| F1$_\text{ref}$ | X | X | 0.156 | 0.253 |

## 5 Related Work

The evaluation of causal discovery algorithms mainly consists of synthetic and real data experiments. Synthetic data are mostly sampled from randomly generated graph structures, or based on models proposed in different works. Such synthetic data experiments can show the superior performance of proposed methods but sometimes may oversimplify the challenges in real-world scenarios [15]. Unfortunately, there are few available real-world datasets for evaluating causal discovery algorithms. Mooij et al. [25] provided a set of cause-effect pairs with ground-truth causal relations. However,

the cause-effect pairs can be used for the evaluation of a limited range of causal discovery methods such as the Linear Non-Gaussian Acyclic Model (LiNGAM) [37]. Also, the dataset containing only pair-wise data is not complex enough to evaluate causal discovery algorithms in real-world scenarios. Several other datasets from genomics [30, 35, 14] and health-care [44] contain causal relations among multiple variables and are commonly used for the evaluation; however, few pairs of ground-true causal relations are known/labeled by domain experts and the evaluation is not systematic. Therefore, it is necessary to develop causal discovery benchmarks for real-world evaluation.

Filling the gap between the synthetic and real data evaluation [17], the simulator in the context of real-world applications is needed. Glymour et al. [17] discussed the evaluation of search tasks, especially causal discovery, and concluded that simulation is a desired way to evaluate the research in this direction. Despite the argument, [17] did not build any simulator instance. Very recently, a few simulators for causal discovery evaluation have been developed, especially considering time-series data. Sanchez-Romero et al. [36] generated simulated fMRI data over time with the focus on the situation where feedback loops exist. Runge et al. [33] provided ground-truth time-series datasets by mimicking properties of real climate and weather datasets. However, these simulators are still limited to the complexity reflecting real-world causal discovery demands and are not suitable for evaluating the causal discovery methods for static data.

In machine learning, there are many simulators built for other disciplines. For example, reinforcement learning benefits from the simulators covering practical issues with different applications [8, 5, 19]. Some of them are used for evaluating sequential decision making by considering counterfactual outcomes. Oberst and Sontag [26] simulated data about treating sepsis among intensive care unit (ICU) patients. The data consist of vital signs, treatment options, and the final mortality with a fully specified underlying Markov Decision Process. Another simulator [16] is used for evaluating the performance of the treatment response over time [21]. Geng et al. [16] provided the dynamics of the tumor volume and its relation with chemotherapy, tumor growth, and radiation. Given parameters of the dynamic equations, Lim [21] simulated the data satisfying this domain knowledge and introduced the practical issues such as unmeasured confounding. However, these simulators contribute to advancing the research on estimating treatment response over time but not causal discovery.

# 6 Discussion

In this work, we build a simulator in the neuropathic pain diagnosis setting for evaluating causal discovery algorithms. Our simulator is based on ground-truth causal relations regarding the domain knowledge, and its parameters are estimated with a real-world dataset. It contains 222 nodes and 770 edges establishing complex real-world challenges. Our simulator can generate any amount of synthetic records that are indistinguishable from real-world records judged by physicians. The simulator can also simulate practical issues in causal discovery research such as missing data, selection bias, and unknown confounding. We demonstrated how to evaluate causal discovery algorithms using our simulator for different challenges.

Our simulator not only contributes to causal discovery research but also machine learning in healthcare research where public data are extremely scarce due to privacy concerns. In the future, we will refine our simulator to consider border scenarios. At the same time, we will seek further opportunities to build different simulators for causal discovery evaluation and machine learning in healthcare research.

**Acknowledgements.** Kun Zhang would like to acknowledge the support by National Institutes of Health under Contract No. NIH-1R01EB022858-01, FAINR01EB022858, NIH-1R01LM012087, NIH-5U54HG008540-02, and FAIN- U54HG008540, by the United States Air Force under Contract No. FA8650-17-C-7715, and by National Science Foundation EAGER Grant No. IIS-1829681. The National Institutes of Health, the U.S. Air Force, and the National Science Foundation are not responsible for the views reported in this article. In addition, the authors thank Akshaya Thippur Sridatta and Tino Weinkauf for the help of the audio dubbing of the 3-minute introduction video at `https://youtu.be/1UvVnIbjSX8` and the visualization of the causal graph.

## Footnotes

[1] The simulator is available at https://github.com/TURuibo/Neuropathic-Pain-Diagnosis-Simulator.

[2] The dataset is collected in a hospital department specialized in neuropathic pain [46]. Only Ruibo Tu and Bo C. Bertilson get access to the dataset during the course of the project.

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
