[Supplementary Material]

# A  Ground-truth Causal Relations

In this paper, we focus on the neuropathic pain caused by discoligamentous injuries and radiculopathies. Table 11 shows all the ground-truth causal relations which are used for establishing the ground-truth causal graph for our simulator. Figure 5 shows the whole causal graph of neuropathic pain diagnose.

Figure 5: The causal graph of neuropathic pain diagnoses.

# B  Simulation Quality Evaluation

We show several simulated patient diagnostic records and the corresponding physician's evaluation scores in Table 6. Moreover, we show the correlation matrices of the real dataset and the simulated dataset in Figure 6. We can clearly see the pattern of the correlation matrix of the simulated dataset, which is similar to the pattern of the correlation matrix of the real dataset. The correlation matrix of the simulated dataset contains many white lines. When all values of a variable are zero in the simulated dataset, the row and the column of the variable in the correlation matrix are white lines. Since our simulator cancels out many correlations between variables which might be introduced by unknown confounders and selection bias, the simulated dataset with a small sample size might have many variables with only zero value. The number of full-zero variables can be controlled by introducing different levels of random noise in the data generation process.

Table 6: A part of the physician's evaluation results.

| Pathophysiology diagnosis | Pattern diagnosis | Symptom diagnosis | Score |
|---|---|---|---|
| DLI L3-L4, DLI T11-T12 | L L4 Radiculopathy, R L4 Radiculopathy | L Front thigh pain, R Front thigh pain, R shin | 4 |
| DLI C3-C4, DLI L4-L5, DLI L5-S1 | L C4 Radiculopathy , R C4 Radiculopathy, L L5 Radiculopathy, R L5 Radiculopathy, L S1 Radiculopathy, R S1 Radiculopathy | L Neck problems, R neck, L collarbone pain, R Front Axle Problems, L shoulder impingement, L Shoulder trouble, R Shoulder problems, R Shoulder trouble, L PTA, L Front knee pain, R Front knee pain, R arch, L Obesity, R Ham, R Tear problems, R heel problems, L Heel problems, L rear thigh pain | 4 |
| DLI C2-C3 | L C3 Radiculopathy, R C3 Radiculopathy | L Neck problems, R neck | 3 |
| DLI C1-C2, DLI C5-C6, DLI C6-C7, DLI S1-S2 | L C2 Radiculopathy, R C2 Radiculopathy, L C6 Radiculopathy, R C6 Radiculopathy', L C7 Radiculopathy, R C7 Radiculopathy | neck pain, L Eye problems, R Eye problems, L Jaw problems, L forehead Headache, R Jaw trouble, L Shoulder problems, R Shoulder problems, L Thumbs up, L Hand problems, R Armband, L Medial elbow problems, R Medial elbow problems | 3 |
| DLI L4-L5, DLI L5-S1 | L L5 Radiculopathy, R L5 Radiculopathy, L S1 Radiculopathy, R S1 Radiculopathy | R adductor tendonitis , lumbago, R Lumbago, L Hip joint, R Hip arthritis, L Medial knee joint disorder, R shin, R Knee trouble, R Tear problems, L Lateral Foot Disorders, L Heel problems, R Achilles tendency | 2 |
| DLI L4-L5, DLI L5-S1 | L L5 Radiculopathy, R L5 Radiculopathy, R S1 Radiculopathy | lumbago, L Hip joint, L Ankle trouble, L Footstool trouble, R Dorsal knee joint disorder, Coccydyni, R Rear thigh pain, R Achilles tendency | 2 |

# C  Experiment Details

In this section, we mainly show recall, precision, and SHD results of causal discovery algorithms under different experiment settings. Table 7, Table 8, Table 9, and Table 10 are the results of the experiments designed for evaluating causal discovery algorithms in the presence of different sample sizes, unknown confounders, selection bias, and missing data.

For generating the datasets with confounders that are external variables, we choose the discoligamentous injury C2-C3, C3-C4, C4-C5, C5-C6, C6-C7, and C7-C8 as the direct effects of an unknown confounder. In this experiment, the confounder can be interpreted as the occupation that can easily damage the neck part of people during the work time. Thus, the chosen discoligamentous injuries are correlated with each other. Then, we use Bernoulli distribution as the marginal distribution of the confounder, and assign a default CPD to each chosen discoligamentous injury. In the end, we generate the data from our modified simulator and delete the data of the introduced confounder in the simulated dataset.

(a) Correlation matrix of the real dataset      (b) Correlation matrix of the simulated dataset

Figure 6: Comparison of the correlation matrices of the real and simulated datasets.

Table 7: Performance of different causal discovery algorithms on the datasets with different sample sizes. The SHD performance is better when it has smaller value. Precision, recall, causal accuracy and F1 score are better when they have larger value. Total SHD is 24642.

| Sample size | 128 | 256 | 512 | 1024 | 2048 | 4096 | 8192 | 16384 |
|---|---|---|---|---|---|---|---|---|
| $\text{SHD}_{PC}$ | 794.5 | 809.0 | 830.0 | 834.5 | 826.5 | 822.5 | 801.5 | 802 |
| $\text{SHD}_{GES}$ | 800.5 | 802.5 | 814.0 | 815.0 | 824.0 | 834.0 | 808.5 | 773 |
| $\text{Precision}_{PC}$ | 0.143 | 0.141 | 0.069 | 0.138 | 0.221 | 0.301 | 0.413 | 0.438 |
| $\text{Precision}_{GES}$ | 0.200 | 0.316 | 0.344 | 0.361 | 0.366 | 0.393 | 0.441 | 0.505 |
| $\text{Recall}_{PC}$ | 0.010 | 0.0156 | 0.009 | 0.023 | 0.039 | 0.060 | 0.086 | 0.120 |
| $\text{Recall}_{GES}$ | 0.023 | 0.048 | 0.073 | 0.095 | 0.113 | 0.149 | 0.185 | 0.239 |

Table 8: Performance of causal discovery methods in the presence of unknown confounders. Total SHD is 10082. The sample size is 1024.

| | FCI | RFCI | GFCI | PC | GES |
|---|---|---|---|---|---|
| Cau_Acc | 0.013 | 0.014 | 0.009 | X | X |
| SHD | X | X | X | 123.5 | 83.5 |

Table 11: Ground-truth causal relations. A capital letter with a number represents a radiculopathy. For example, "C2" represents "C2 radiculopathy". A radiculopathy as an effect in the table represents both sides of a radiculopathy. A radiculopathy as a cause in the table has the same side with its effect. We denote left as "L" and right as "R".

| Effect | Cause |
|---|---|
| C2 | DLS C1-C2 |
| C3 | DLS C2-C3 |
| C4 | DLS C3-C4 |
| C5 | DLS C4-C5 |
| C6 | DLS C5-C6 |
| C7 | DLS C6-C7 |
| C8 | DLS C7-C8 |
| T1 | DLS C8-T1 |
| T2 | DLS T1-T2 |
| T3 | DLS T2-T3 |
| T4 | DLS T3-T4 |
| T5 | DLS T4-T5 |
| T6 | DLS T5-T6 |
| T7 | DLS T6-T7 |
| T8 | DLS T7-T8 |
| T9 | DLS T8-T9 |
| T10 | DLS T9-T10 |
| T11 | DLS T10-T11 |

| | |
|---|---|
| T12 | DLS T11-T12 |
| L1 | DLS T12-L1 |
| L2 | DLS L1-L2 |
| L3 | DLS L2-L3 |
| L4 | DLS L3-L4 |
| L5 | DLS L4-L5 |
| S1 | DLS L5-S1 |
| S2 | DLS S1-S2 |
| C2 | craniocervical junction |
| C3 | craniocervical junction |
| C4 | craniocervical junction |
| lbs | T10;T11;T12;L1;L2 |
| L neck problems | C2;C3;C4;C5;C6;C7 |
| Neck pain | C2;C3;C4;C5;C6;C7 |
| R neck | C2;C3;C4;C5;C6;C7 |
| L tinnitus | C2 |
| L eye problems | C2 |
| L ear problems | C2 |
| R tinnitus | C2 |
| R eye problems | C2 |
| R ear problems | C2 |
| Headache | C2 |
| L jaw problems | C2 |
| L forehead headache | C2;C3 |
| Mouth | C2;C3 |
| Forehead headache | C2;C3 |
| R headache | C2;C3 |
| R pta | L4;L5 |
| Pharyngeal discomfort | C2;C3 |
| R jaw trouble | C2;C3 |
| Back headache | C3 |
| R back headache pain | C3 |
| L collarbone pain | C3;C4 |
| R collarbone problems | C3;C4 |
| Central chest pain | C3;C4 |
| L central chest pain | C3;C4 |
| L central chest disorders | C3;C4 |
| R front axle problems | C4;C5;C6 |
| L shoulder impingement | C4;C5;C6 |
| R shoulder impingement | C4;C5;C6 |
| L shoulder problems | C4;C5;C6;C7;C8 |
| L shoulder trouble | C4;C5;C6;C7;C8 |
| R shoulder problems | C4;C5;C6;C7;C8 |
| R shoulder trouble | C4;C5;C6;C7;C8 |
| L upper arm discomfort | C5 |
| L upper elbow pain | C5 |
| Intracapular problems | C5;C6 |
| L interscapular complaints | C5;C6 |
| R intracapular trouble | C5;C6 |
| L lateral elbow pain | C5;C6 |
| L lateral arm discomfort | C5;C6 |
| R lateral elbow pain | C5;C6 |
| L elbow problems | C5;C6;C7;C8 |
| R elbow trouble | C5;C6;C7;C8 |
| L arm | C5;C6;C7;C8;T1 |
| L thumbs up | C6 |
| R thumbs up | C6 |
| L wrist problems | C6;C7 |
| R wrist problems | C6;C7 |

| | |
|---|---|
| L lower arm disorders | C6;C7;C8 |
| R lower arm disorders | C6;C7;C8 |
| L hand problems | C6;C7;C8 |
| R hand problems | C6;C7;C8 |
| L bend of arm problems | C6;C7;C8;T1 |
| R armband | C6;C7;C8;T1 |
| R bend of arm discomfort | C6;C7;C8;T1 |
| L medial elbow problems | C7;C8 |
| R medial elbow problems | C7;C8 |
| L finger trouble | C7;C8 |
| R finger trouble | C7;C8 |
| L small finger trouble | C8 |
| R little finger trouble | C8 |
| L groin trouble | L1;L2 |
| L medial groin disorders | L1;L2 |
| L lateral groin discomfort | L1;L2 |
| Central groin disorders | L1;L2 |
| R lateral groin discomfort | L1;L2 |
| R groin trouble | L1;L2 |
| L adductor tendon | L1;L2;S1;S2 |
| R adductor tendonitis | L1;L2;S1;S2 |
| L hip disorders | L2;L3 |
| L backache | L2;L3;L4;L5;S1 |
| Backache | L2;L3;L4;L5;S1 |
| L lumbago | L2;L3;L4;L5;S1 |
| Lumbago | L2;L3;L4;L5;S1 |
| R lumbago | L2;L3;L4;L5;S1 |
| L front thigh pain | L3;L4 |
| R front thigh pain | L3;L4 |
| R thigh problems | L3;L4;L5;S1 |
| L leg problems | L3;L4;L5;S1 |
| L thigh pain | L3;L4;L5;S1 |
| R leg problems | L3;L4;L5;S1 |
| R medial vadbesvär | L4 |
| L pta | L4;L5 |
| L hip joint | L4;L5 |
| R hip trouble | L4;L5 |
| R hip arthritis | L4;L5 |
| L medial knee joint disorder | L4;L5 |
| L front knee pain | L4;L5 |
| R medial knee joint disorder | L4;L5 |
| R front knee pain | L4;L5 |
| L shin | L4;L5 |
| R shin | L4;L5 |
| L llower leg problems | L4;L5;S1 |
| L knee trouble | L4;L5;S1 |
| R knee trouble | L4;L5;S1 |
| L tåledbesvär | L5 |
| L big toe problems | L5 |
| R big toe problems | L5 |
| L foot pain | L5 |
| L ankle trouble | L5 |
| R ankle trouble | L5 |
| L footstool trouble | L5 |
| R arch | L5 |
| R morton trouble | L5 |
| R fainting | L5 |
| L ischias | L5;S1;S2 |
| R ischias | L5;S1;S2 |

| | |
|---|---|
| L ham | L5;S1 |
| L obesity | L5;S1 |
| R ham | L5;S1 |
| L toe problems | L5;S1 |
| R foot pain | L5;S1 |
| R tear problems | L5;S1 |
| R obesity | L5;S1 |
| R dorsal knee joint disorder | S1 |
| L dorsal knee joint disorder | S1 |
| L lateral knee pain | S1 |
| R lateral knee pain | S1 |
| L small toe trouble | S1 |
| L lateral foot disorders | S1 |
| R lateral foot disorders | S1 |
| R heel problems | S1 |
| Calcaneal pain | S1 |
| L heel problems | S1 |
| Coccydyni | S1 |
| L rear thigh pain | S1 |
| R rear thigh pain | S1 |
| L achilles problems | S1 |
| L achilles tendon | S1 |
| L achillodyni | S1 |
| R achilles problems | S1 |
| R achilles tendency | S1 |
| R achillodyni | S1 |
| Breast backache | T1;T2;T3;T4;T5;T6;T7;T8;T9;T10 |
| Chest discomfort | T3;T4;T5 |
| L breast problems | T3;T4;T5 |
| R breast problems | T3;T4;T5 |
| Toracal dysfunction | T3;T4;T5;T6;T7 |
| Upper abdominal discomfort | T6;T7;T8 |
| Lateral abdominal discomfort | T6;T7;T8;T9;T10;T11;T12;L1;L2 |
| Abdominal discomfort | T6;T7;T8;T9;T10;T11;T12;L1;L2 |
| L lower abdominal discomfort | T9;T10;T11;T12;L1;L2 |
| Lower abdominal discomfort | T9;T10;T11;T12;L1;L2 |

Table 9: Performance of different causal discovery methods in the presence of selection bias.

|  | F1 | F1 ref | Recall | Recall ref | Precision | Precision ref |
|---|---|---|---|---|---|---|
| PC | 0.072 | 0.076 | 0.042 | 0.045 | 0.276 | 0.236 |
| GES | 0.193 | 0.185 | 0.126 | 0.125 | 0.416 | 0.356 |

Table 10: Performance of causal discovery methods in the presence of missing data.

|  | FCI | RFCI | PC | GES |
|---|---|---|---|---|
| $\text{CauAcc}_{\text{MNAR}}$ | 0.059 | 0.051 | 0.061 | 0.154 |
| $\text{CauAcc}_{\text{MAR}}$ | 0.063 | 0.049 | 0.050 | 0.135 |
| $\text{CauAcc}_{\text{MCAR}}$ | 0.066 | 0.055 | 0.067 | 0.161 |
| $\text{CauAcc}_{\text{ref}}$ | 0.062 | 0.050 | 0.059 | 0.145 |
| $\text{SHD}_{\text{MNAR}}$ | X | X | 806.5 | 812.0 |
| $\text{SHD}_{\text{MAR}}$ | X | X | 806.5 | 778.5 |
| $\text{SHD}_{\text{MCAR}}$ | X | X | 804.5 | 801.5 |
| $\text{SHD}_{\text{ref}}$ | X | X | 795.0 | 765.5 |
| $\text{Recall}_{\text{MNAR}}$ | X | X | 0.081 | 0.177 |
| $\text{Recall}_{\text{MAR}}$ | X | X | 0.080 | 0.160 |
| $\text{Recall}_{\text{MCAR}}$ | X | X | 0.086 | 0.181 |
| $\text{Recall}_{\text{ref}}$ | X | X | 0.094 | 0.168 |
| $\text{Precision}_{\text{MNAR}}$ | X | X | 0.376 | 0.435 |
| $\text{Precision}_{\text{MAR}}$ | X | X | 0.389 | 0.490 |
| $\text{Precision}_{\text{MCAR}}$ | X | X | 0.405 | 0.440 |
| $\text{Precision}_{\text{ref}}$ | X | X | 0.462 | 0.514 |
| $\text{F1}_{\text{MNAR}}$ | X | X | 0.133 | 0.251 |
| $\text{F1}_{\text{MAR}}$ | X | X | 0.132 | 0.241 |
| $\text{F1}_{\text{MCAR}}$ | X | X | 0.141 | 0.256 |
| $\text{F1}_{\text{ref}}$ | X | X | 0.156 | 0.253 |