[Reviews · NeurIPS 2019]

Reviewer 1



The authors propose a neuropathic pain simulator to generate data for causal discovery algorithm evaluation. To the best of my knowledge, this is the first work to build a simulator that can generate data with causal relations. This paper is novel, but my concern is that it is not rigorous enough. To make the simulated data close to the real-world scenario, the authors learn the simulator from a real-world dataset. However, the real-world dataset only contains 141 patient diagnostic records. I am not sure whether this amount of data includes enough causal relations for estimating the conditional probability distribution of each variable given other variables. If not, the simulator may generate data with biases. The section 2.2 does not provide a clear statement. The authors should first formally introduce the format of the real-word dataset and then present the procedure of the parameter estimation. In addition, there exists a potential problem about the heuristic proposed in this section. The heuristic indicates that if a parent Pa1(X) and another parent Pa2(X) of a variable X happen at the same time, the conditional probability P(X=1|Pa1(X)=1,Pa2(X)=1) is higher than or equal to the maximum value of P(X=1|Pa1(X)=1) and P(X=1|Pa2(X)=1). The authors only focus on the causal relations between that the parent value is 1 (Pai(X)=1) and a variable X. What if the parent of X having value 0 (Pai(X)=0) has causal relations with a variable X? Although the experimental results show that the physicians cannot differentiate the simulated data from the real-world data, I think the experiment may exist some sampling biases. The size of experimental dataset used in the physician evaluation is too small. To examine the quality of simulated data, the authors only mix 30 simulated records with 30 data sampled from the real-world dataset. This sampling bias can highly affect the evaluation results. The causal relation considered in this paper is relatively simple. In this paper, the causal relation consists of three layers: pathophysiological diagnosis, pattern diagnosis, and symptom diagnosis. The symptom diagnosis is only caused by the pattern diagnosis, and the pattern diagnosis is only caused by the pathophysiological diagnosis. Moreover, the nodes in each layer have no causal relation with each other. In the real world, the causal relation is much more complex than the setting presented in the paper. If the simulator can generate dataset with complex causal relations, it can be a better benchmark.

Reviewer 2



1. Thanks to the known surface map of the region controlled by different nerves and empirically known relationship between various diseases and corresponding pain regions it is possible to reconstruct a complete graph of relations for this small subdomain. The paper proposes a causal simulator with this graph structure and parameters estimated based on real data example. The samples from the simulator are evaluated by human experts and concluded to be more likely indicators of known diseases than the real data. To demonstrate the utility of the simulator the paper runs a comparison of a few well established causal learning algorithms. The general idea is great and the authors should definitely place their simulator online and actively advertise is to the interested group of researchers. However, the binary nature of the data and a highly specific structure of the causal graph limit the application area of the simulator. I am not sure if NeurIPS is the right venue for presenting the simulator although I understand the paradoxical feel of this statement. Yes causal learning needs the tools like presented in order to simplify development. However, I feel a paper where a proposed simulator would have been used to simplify development of a novel algorithm and demonstrate that the thus obtained approach beats the state of the art would be more appropriate. Note, I fully support the proposed work and my only concern is the fit. 2. It would be nice to see a layout of the complete graph as the partial representation in Figure 2, which currently is a tree and does not convey the sense of possible complexities of the proposed model graph. 3. With the BDeU score, GES is naturally fit for the binary data produced by the simulator but it is unclear what test was used for the PC and FCI algorithms.

Reviewer 3



Originality: In terms of causal inference, the paper is highly original, as there are few works devoted to developing new simulation systems that can provide realistic evaluations. However, the related work section is somewhat sparse and it's not clear to me how novel the system is in terms of the application domain. In general, the related work could be improved. Quality: The paper clearly articulates a gap in the literature, introduces a seemingly sound simulation system, and presents well thought out evaluations. I was especially pleased to see evaluation with domain experts. Clarity: The paper is very well written, and the methods are well-reasoned. Some questions: -It is not clear to me why there are ~800 causal relationships. In medical records I am familiar with diagnosis codes such as ICD9/10 are used. Since ICD codes form a tree, one can choose highly specific or general codes for the same event. Potentially if one used the most specific code for every illness, this would lead to many relationships (though many may be redundant, if for example every child of a node shares the same effect). Is something similar at play here? -For section 2.3, there is minimal detail about how these components are implemented. Are they part of the simulation system that will be made available, or something the authors did in post-processing of the data generated?

[Author Response · NeurIPS 2019]

We thank all reviewers for the constructive feedback. We address all concerns below.

**Response to Reviewer 1** » not rigorous enough... the real-world dataset size ...
biases ... physician evaluation size.

Figure 1: New physician's evaluation results of 100 data examples.

Due to the costly nature of medical data sets, the original data set size is not big, as
you pointed out. However, we note that this is in fact a strong motivation to build
such a simulator. Also, as you mentioned, this is the first work to build a simulator
that can generate data with causal relations–the data set size will naturally grow
in the future and this work will hopefully inspire improved simulation systems.
Moreover, a small data set size does not seem necessary to introduce bias. (Here
by "bias," we mean bias in the estimated parameter values.) The real-world data
set is collected by using two months of clinical records. Our simulation data
satisfy distribution properties of the real-world data set. This is further suggested by both of the updated and original
physician's evaluations.

Regarding the physician evaluation, the samples given to the physician for evaluation were also sampled uniformly,
which is unbiased. Also, we have added another 40 examples (20 simulation records and 20 real-world records), which
increased the physician evaluation from 60 examples to 100 examples. The new results with 100 examples are shown in
Figure 1. We find that the physician evaluation using 100 samples is consistent with the evaluation using 60 samples.
Both of the updated and original evaluation figures in the paper indicate that our simulator simulates realistic patient
diagnosis records where the physician cannot distinguish from real-life cases.

» The section 2.2... first the format of the real-world dataset. Thanks for the feedback. We will adjust the order and
introduce the format of the data set in the beginning of Section 2.2.

» ... the heuristic ... It is not heuristic, but rigorous. We clarify it below and also in the updated version of the paper.
Please note that components of **c** in equation (1) can be 1 or 0 (they are not $c_i$, which is 1); see the line below equation
(1). In words, our heuristic estimation considers the case where parents of a variable having value 0, e.g., we estimate
$P(X = 1 \mid Pa_1(X) = 0, Pa_2(X) = 1)$ with $P(X = 1 \mid Pa_2(X) = 1)$ which is also supported by the medical
insight that $P(X = 1 \mid Pa_1(X) = 0, Pa_2(X) = 1) \leq P(X = 1 \mid Pa_2(X) = 1)$.

» The causal relation is relatively simple ... three layers ... In our system, the complexity of the causal relationship
is mainly reflected in the coverage of different types of causal structures rather than the depth of the longest chain.
Our causal graph covers all the basic d-separation relations, i.e., the chain structure, the collider structure, and the
common cause structure. Additionally, domain knowledge and physician's experiences may reduce the complexity of
the ground-true causal graph structure, but it will not reduce the complexity of the task, which recovers the true causal
graph structure from a complete/empty graph.

» ... broader scenarios ... not just neuropathic pain... As you pointed out, this work is the first step towards this goal
of systematic causal discovery evaluation. We hope and believe that this work will inspire the development of other
simulators with various types of causal relationships.

**Response to Reviewer 2** Thanks for your encouraging comments. » ... venue ... Thanks for sharing your concern.
Your proposal is great. At the same time, because of the urgent need of simulators for causal discovery method
evaluation, we believe that this work will benefit a large range of causality researchers in the NeurIPS community.

» ... a layout of the complete graph ... We have presented the complete graph in the form of cause-effect node pairs in
the appendix. Due to the size of the graph, we will visualize it using an online graph visualization tool and attach the
link to the updated version of the paper.

» ... test for the PC and FCI ... For PC and FCI we used Chi-squared test. We will make it explicit.

**Response to Reviewer 3** » ... around 800 causal relationships ... In this work, we directly used the diagnostic terms
from physicians instead of codes such as ICD codes. Also, we differ the left and right side for both symptom diagnostic
terms and pattern diagnostic terms. This leads to around 200 nodes and 800 edges.

» ... implementation ... We will open source the entire simulator, including the ground-truth causal graph, the estimated
parameters, the simulator for data generation, and examples under different scenarios shown in the paper.

» ... generalizability ... time series. Although our simulator has already covered many practical challenges in the causal
discovery domain, we agree with the reviewer that it would be of higher impact to further generalize it, for example, to
time series. This is feasible but needs some real-world time-series data to support the parameter estimation. We are
working closely with medical doctors to enlarge the cohort, and generalizing it to consider time series is one of our
future work direction.

[Meta-Review · NeurIPS 2019]

The authors propose a simulator that can generate benchmark datasets with causal relations based on patient diagnostic records. Expert from the field were involved to validate the simulator. Even though the simulations are moderately complex the simulator might be helpful to investigate causal inference methods. An empirical study compares few existing algorithms. For this purpose the simulator is distributed as python package.